# Application of a 3D-Printed Part with Conformal Cooling in High-Pressure Die Casting Mould and Evaluation of Stress State During Exploitation

**DOI:** 10.3390/ma17235988

**Published:** 2024-12-06

**Authors:** Marcin Małysza, Robert Żuczek, Dorota Wilk-Kołodziejczyk, Krzysztof Jaśkowiec, Adam Bitka, Mirosław Głowacki, Łukasz Zięba, Stanisław Pysz

**Affiliations:** 1Faculty of Metals Engineering and Industrial Computer Science, AGH University of Krakow, al. Mickiewicza 30, 30-059 Krakow, Poland; marcin.malysza@kit.lukasiewicz.gov.pl (M.M.); krzysztof.jaskowiec@kit.lukasiewicz.gov.pl (K.J.); glowacki@agh.edu.pl (M.G.); 2Łukasiewicz Research Network—Krakow Institute of Technology, 30-418 Krakow, Poland; robert.zuczek@kit.lukasiewicz.gov.pl (R.Ż.); adam.bitka@kit.lukasiewicz.gov.pl (A.B.); 3Faculty of Foundry, AGH University of Krakow, al. Mickiewicza 30, 30-059 Krakow, Poland; 4Faculty of Natural Sciences, Jan Kochanowski University of Kielce, ul. Żeromskiego, 25-369 Kielce, Poland; 5ARP Radom, ul. Tarnobrzeska 9, 26-600 Radom, Poland; lukasz.zieba@arp.pl

**Keywords:** 3D printing, high pressure die casting, stress analysis, computer simulation

## Abstract

The article addresses stress formation in the structural 3D-printed elements of a high-pressure die casting die mould used for production of aluminum castings. The 3D-printed elements with conformal cooling are manufactured of 18Ni300 powder. Initial numerical calculations were performed on a test die mould made of standard steel X40CrMoV5 to determine temperature distribution and stress state, providing a baseline for comparing 3D-printed 18Ni300 parts. A database for 18Ni300 material was developed, including optimal heat treatment parameters: aged at 560 °C for 8 h. The resulting tensile strength of approximately ~1600 MPa, yield strength 1550 MPa, and elongation 6–7%, with properties temperature-dependent from 20 °C to 600 °C. Results show that conformal cooling increases stress gradients, highlighting the demands on fatigue strength at elevated temperatures. The study revealed that the heat treatment significantly influences the final properties, with tensile strengths of 1400–2000 MPa and elongation from 1 to 8%. While the heat treatment has a greater impact on the mechanical properties than the printing parameters, optimizing the printing settings remains crucial for ensuring density and quality in the die moulds under cyclic loads.

## 1. Introduction

The application of Additive Manufacturing (AM) techniques for producing inserts in die-casting moulds has been a focus of research for nearly 20 years. This interest arises from AM’s potential to create advanced cooling systems (conformal cooling) that reduce casting defects and shorten production cycles. While AM was initially more common in plastic injection moulding, where thermal loads are lower, recent advancements in equipment and metal powder technologies have made it increasingly viable in die-casting. Among the materials explored, 18Ni300 (maraging steel—1.2709) is widely used in moulds exposed to variable temperature gradients due to its excellent properties, such as a tensile strength of Rm > 1800 MPa, hardness HRC > 50, and elongation A5 > 8%. Maraging steels, which are produced under precise vacuum conditions using VIM (Vacuum Induction Melting) and VAR (Vacuum Arc Remelting), followed by plastic deformation and heat treatment, are known for high crack resistance, minimal distortion during heat treatment, and ease of welding. The existing literature on AM with maraging metal powder has largely focused on understanding the relationships between printing parameters, heat treatments, and resulting microstructures—primarily the formation of precipitates and their role in strengthening. However, few studies address the macrostructural behavior of printed components subjected to variable thermal loads, such as die-casting inserts, or provide specific guidelines for optimizing printing and heat treatment processes for 18Ni300 powder. This study seeks to fill the gap by investigating whether AM-produced inserts can achieve the necessary mechanical properties—specifically, tensile strength and hardness comparable to those of conventional tool steels like Uddeholm Dievar, which offers a tensile strength Rm above 1600 MPa, hardness between 45 and 55 HRC, and elongation around 10%. While AM-produced inserts reach similar tensile strength and hardness, their elongation typically falls short, at only 1–3%. This discrepancy raises questions about the potential of AM-produced inserts to withstand the demanding conditions of die-casting moulds and whether plasticity can be improved through careful control of printing and heat treatment parameters. To address this, our research examined a range of variables impacting plastic properties, including powder characteristics, laser power, feed rate, layer thickness, heat treatment temperature and duration, and post-process adjustments. The unique microstructure achieved through AM, primarily martensitic with retained austenite due to rapid cooling rates (~10^5^ K/s), differs significantly from structures obtained via traditional casting or forging. Additionally, AM introduces distinct defects (such as microporosity from shrinkage, gas entrapment, and incomplete melting), which can compromise the final properties of the printed components. The study involved extensive experimentation to determine the optimal parameters for the AM process on a selected 3D-printing device, assessing how different heat treatment regimens influence the microstructure and mechanical properties of printed samples. Mechanical testing of samples produced with various process parameters was performed, followed by numerical analysis to design and validate a component for die-casting moulds that ensures reliable performance under variable thermal loads. Through this work, we aim to provide valuable insights and process guidelines for producing durable AM inserts with conformal cooling for die-casting moulds [1,2,3,4,5,6,7,8,9].

## 2. Materials and Methods

The presented study was planned to be performed in a few steps. First of all was to determine the manufacturing procedure of 3D elements with material 18Ni300 (maraging steel—1.2709). Regarding the results of 3D-printing procedure, the porosity occurrence was determined. Next for the best process parameters, the heat treatment procedure was set for different time-temperature steps. For that the Differential Scanning Calorimetry Technique was used. Based on the DSC results the thermodynamic phenomenon that occurs during temperature changes was acquired. Along with the laboratory trials, the virtual experiments regarding the operation of HPDC die mould were determined. For that, the variable temperature field and stresses during the die-casting mould cycle, numerical calculations were performed using MagmaSoft 5.2 and Flow3D Cast 2023R2 software. The calculations were based on a selected technology for casting a test model of a sprue bushing as a component of a die-casting mould. The study of the impact of printing parameters on properties was conducted on samples obtained using the Laser Metal Deposition (LMD) technology on an RPMI 557 machine (RPM Innovations, Inc., Rapid City, SD, USA) (Figure 1).

The selection of printing parameters was made by evaluating the energy density for which the lowest porosity value was obtained. The energy density combines laser power with the spot’s travel speed, layer thickness, and track width, and is expressed by the formula:E = P/(∂ × V × h)
where

E—energy density, J/mm^3^

P—laser power, W

V—scanning speed, mm/s

δ—layer thickness, mm

h—distance between tracks, mm

Porosity was analyzed using three methods: the ImageJ 1.54 program from metallographic microsection images (NIKON Eclipse LV100PO, Healthcare Business Unit, Shinagawa Intercity Tower C, Tokyo, Japan), density measurements by the Archimedes method, and computed tomography using a “Nanotom” CT scanner (Wydział Mechaniczny Politechniki Krakowskiej, Kraków, Poland). Metallographic samples for porosity evaluation were ground on abrasive papers with a gradation of 220 µm, then polished on polishing cloths using diamond pastes with a gradation from 9 to 0.25 µm. The microstructure was evaluated on sections etched with Mi1Fe reagent (Nital). Microstructure and chemical composition studies were performed using an Axio Observer Z1m light microscope (Carl Zeiss Jena GmbH, Jena, Germany) and a Scios FEG, FEI scanning electron microscope. Local elemental composition analysis (EDS) and phase component identification using EBSD were also conducted. Microscopic observations and photographs were taken with the SCIOS-FEG, FEI scanning electron microscope. Local chemical composition analysis (EDS) in a selected microarea was performed using an EDS, EDAX X-ray microanalyzer (AMETEK GmbH, Weiterstadt, Germany). Phase component identification was carried out using backscattered electron diffraction (EBSD). Microhardness tests were conducted on a multifunctional measurement platform for mechanical properties testing on a micro scale, Anton Paar (Warszawa, Poland). Microhardness tests were performed using a Vickers indenter (Struers S.A.S., Champigny sur Marne, France) at a load of HV0.05. Static strength tests at room temperature and at 200 °C, 400 °C, and 600 °C were performed on an INSTRON 8800M machine (Norwood, MA, USA) with velocity of 1 mm/min, the heating of the sample was introduced in Maytec HTO-09 (MayTec Aluminium Systemtechnik GmbH, Olching, Germany)—temperature range 0–1200 °C, the static test for sample B shape Zwick Roell Z010 machine (ZwickRoell Sp. z o.o. Sp. k., Wroclaw, Poland) was used with effective traverse velocity of 0.0283 mm/s, velocity of determining the yield strength 30 MPa/s (Figure 2).

## 3. Results

### 3.1. Research and Determination of the Properties of 1.2709 (Maraging) Steel Obtained Through Additive Manufacturing Technology Necessary for Numerical Calculations

#### 3.1.1. Determination of Printer Parameters to Achieve Minimal Porosity

The laser beam printing process, through overlapping paths both horizontally (laser movement) and vertically (subsequent powder layers), generates discontinuities in the material’s structure, leading to the formation of microporosity:shrinkage—transition from liquid to solid phase,gaseous—entrapment of shielding gas as bubbles,incompletely melted powder.

In the conducted research, all types of discontinuities were encountered, as shown in Figure 3. The intensity and size of discontinuities in the structure are related to the quality of the powder, the printing method, and the type of printer used for production. Therefore, it becomes necessary to first determine the optimal process parameters for the powder and printer used in this research. The adopted evaluation criterion was the amount of porosity. Samples were made using an RPMI printer, which uses the LMD—Laser Metal Deposition method. Figure 3 and Figure 4 clearly show the microporosity in the metallographic sections and CT scan images generated during printing with different parameters. The research results are presented in Figure 5. The lowest porosity was achieved for a laser energy density of 1.717 J/mm^3^. The achieved laser density is characterized by power of 1300 W, track width of 1.52 mm, and powder layer height of 0.5 mm. Many articles justify and explain the impact of laser power and scanning speed on porosity. It seems that the selected results reflect similar studies described in these articles. Figure 6 presents the summary of the results of the porosity study along with the energy density.

#### 3.1.2. Determination of Heat Treatment Parameters

Determination of heat treatment parameters for the assumed properties of hot-working steel obtained through additive manufacturing technology. After developing the printing parameters, materials were prepared for further research aimed at selecting heat treatment that allows achieving the best properties of steel printed from 18Ni300 powder. The reference material for determining the properties to be obtained was Uddeholm Dievar steel, commonly used for making components in die-casting moulds operating under varying temperature fields. The research on the strength of Dievar steel after heat treatment and data obtained from facilities using this steel allowed for determining the following parameters that should be achieved for steel printed from 18Ni300 powder:average hardness: 46 ± 2 HRC,tensile strength (Rm): approximately 1600 ± 50 MPa,elongation (A5): approximately 10 ± 1.5%.

The samples made from 18Ni300 powder and examined immediately after printing without heat treatment show the following properties:average hardness: 40 to 43 HRC,tensile strength (Rm): approximately 1050 ± 50 MPa,elongation (A5): approximately 2.5 ± 2%.

The obtained properties indicate low elongation and unsatisfactory strength. Therefore, determining heat treatment parameters is crucial for achieving the desired properties of the material intended for die-casting mould components. Steel’s resistance to thermal cracking is essentially determined by its thermal strength and plasticity. Thermal resistance is directly related to hardness. An increase in hardness increases thermal strength but reduces plasticity (ductility). The mechanical properties (strength, elongation, hardness, impact resistance) of the 1.2709 alloy obtained through printing are a function of aging temperature and time, as these parameters affect the kinetics of intermetallic phase formation, retained austenite growth, and the amount of so-called reversed austenite [10,11]. DSC (Differential Scanning Calorimetry) characteristics show endothermic and exothermic phenomena of materials during heating, related to phase transformation. This allows for a preliminary assessment and adoption of heat treatment parameters. The phase transformation characteristics of a sample directly after printing without heat treatment are shown in Figure 7. The first exothermic peak at 475 °C is associated with carbide formation. The process occurring between 500 °C and 565 °C, where intermetallic phases such as Ni_2_(Ti,Mo), Fe_7_Mo_6_, or Fe_2_Mo form, significantly affects the final mechanical properties. This process increases material strength. Simultaneously, there is growth in retained austenite grain size, formed during printing, and reversed austenite from martensite. Further temperature increases above 600 °C lead to the transformation of martensite into austenite through diffusion, with the maximum intensity related to the first endothermic peak at 660 °C.

DSC studies enabled an approximate determination of the temperature range from 475 °C to about 565 °C, where the intensity of intermetallic phase precipitation increases strength. Simultaneously, increasing plasticity through reversed austenite formation should involve heat treatment around 600 °C. Introducing reversed austenite is a method to increase the plasticity of martensitic steels, including maraging steels. The assumed strength of about 1600 MPa and elongation of approximately 9% must be a compromise of aging temperature and time. Therefore, a research plan on the influence of heat treatment on properties was developed and presented in Table 1.

The influence of austenite on material elongation was confirmed through detailed analyses using the Electron Backscatter Diffraction (EBSD) method on structures formed in samples subjected to various heat treatments, particularly aging at 490 °C for 4 h and 560 °C for 8 h. These studies, combined with mechanical property tests, demonstrated a clear correlation between increased austenite content and improved ductility. The austenite content rose from approximately 10% in samples aged at 490 °C to about 14% in samples aged at 560 °C. This increase was associated with a significant rise in elongation, which grew from 2% to 6%, as shown in Figure 8.

Additionally, samples subjected to different heat treatment variants exhibited varied microstructural characteristics directly related to the amount of austenite formed. Longer aging times and higher temperatures favored the formation of austenite in the form of needle-shaped precipitates. This structural feature was especially pronounced in samples treated at 600 °C for 8 h, where precipitates were clearly visible, as shown in Figure 9. The results of this study suggest that the application of higher aging temperatures and prolonged aging times causes a decrease in material strength while simultaneously increasing its elongation. This relationship is indicative of enhanced ductility achieved through the formation of austenite. In contrast, lower aging temperatures promote higher strength but at the expense of elongation. This relationship between heat treatment parameters and mechanical properties is detailed in Figure 10 which illustrates the opposing effects of different aging temperatures on strength and ductility. The findings provide valuable insights into the potential to tailor the mechanical properties of materials by precisely controlling the heat treatment process, allowing for the adjustment of austenite content to meet specific performance requirements. This confirms the dependence of elongation on strength as presented in Figure 10. Nevertheless, the mechanical properties of the material as a function of temperature are crucial since the goal is to use the printed material for high-temperature work. Steel more resistant to thermal loads should possess high thermal strength and good plasticity (ductility) within the working temperature ranges of the moulds. It can be assumed that the cyclic temperature changes in the mould range from 180 °C to locally 600 °C. Therefore, within these temperature ranges, the steel used for the mould should have the best properties, i.e., high strength (hardness) and very good plasticity. Unfortunately, in most steels, the relationship between strength (hardness) and plasticity is inversely proportional. Therefore, capturing the appropriate relationship of these properties as a function of temperature for a given steel grade is essential. The mechanical strength studies of samples printed from 18Ni300 powder with various heat treatment parameters and tested at 20 °C, 200 °C, and 400 °C are presented in Figure 10, Figure 11, Figure 12 and Figure 13. The strength at 400 °C is around 1310 MPa, decreasing by 19%, while elongation is around 10%, increasing by about 40%. The achieved value of about 1300 MPa at 400 °C is a very good result comparable to hot-working rolled steels. One of the best Swedish hot-working steels, often used for die-casting mould inserts—Dievar, reaches around 1200 MPa at 400 °C, with an elongation of 13%.

The presented studies suggest the best mechanical properties at 20 °C, 200 °C, and 400 °C are achieved with 18Ni300 steel printed and aged at 560 °C for 8 h, similar to those of Dievar steel. The proposed heat treatment parameters are suitable for creating printed materials for the production of die-casting mould components operating under varying temperature fields. In the conducted studies, it was not possible to achieve the assumed elongation of 9% at room temperature. The maximum achieved elongation is 8%, but the strength drops below 1400 MPa (Figure 11). The assumed strength of 1600 MPa was obtained in several heat treatment variants, but only one provided a satisfactory elongation of about 6%. For further research, to develop a database of material made from 18Ni300 powder, the following heat treatment was selected: aging at 560 °C for 8 h.

#### 3.1.3. Development of a Database of Thermophysical Properties for the MAGMA5 Program

The commonly used steel for making elements in pressure die and available in the MAGMA database is the hot-working construction steel X40CrMoV5_1 (according to DIN 17350, ASTM, H13). Therefore, this material was adopted as a reference material during numerical calculations with which the stress state in the die was compared. The thermophysical data for this steel are contained in the MAGMA database and were used in the study. Table 2 presents the chemical composition of X40CrMoV5_1 steel and Dievar, which was previously studied for comparison of its properties with printed samples from 18Ni300 powder.

Both hot-working steels have similar compositions, and their thermophysical properties will be characterized by similar values. Using the JmatPro program and our own research, a database of the 1.2709 alloys obtained in the process of printing from 18Ni300 powder was developed. So far, this material has been tested for mechanical properties such as static strength, elongation, static fatigue, or thermal fatigue. However, performing numerical calculations requires full knowledge of the material properties, including determining the coefficients of thermal expansion, thermal conductivity, or specific heat as a function of temperature, which are necessary to perform numerical calculations.

Table 3 presents some thermophysical parameters and mechanical properties as a function of temperature for both materials used in the calculations.

### 3.2. Stress Analysis in a Working Die Made of Hot-Working Construction Steel X40CrMoV5_1

To answer the question of the correct relationship between strength and elongation in structural elements working in a variable temperature field in pressure moulds, an analysis of the stress state of a working die made of X40CrMoV5_1 was first conducted. Based on this, it will be possible to determine the requirements that an element made using additive manufacturing technology must meet.

#### 3.2.1. The HPDC Die Mould During Operation

The die-casting mould undergoes cyclic temperature changes during operation. Figure 14 shows the temperature distribution at a selected point. During a cycle that lasts about 150 s, there is a rapid temperature increase from 150 °C to 520 °C in 50 s, followed by a drop to 150 °C in 100 s. Such a cycle can repeat approximately 110 times within 8 h of work. Direct contact of liquid metal with the mould surface during injection and solidification (Figure 14) generates a temperature gradient several millimeters deep from the mould surface. At a distance of 5 mm from the surface, the temperature difference is between 20 and 30 °C (Figure 15). Due to these temperature gradients, significant stresses are generated near the surface, repeating with the mould’s cyclic operation. Reduced Von Mises stresses exceed 400 MPa (Figure 16a). Near the surface, the metal structure of the mould is subjected to compressive and tensile stresses (Figure 16b). On the surface itself, compressive stresses dominate and vary throughout the operation cycle. In different areas of the mould’s structure, especially near the gating system, maximum stresses may exceed the allowable yield strength of the material at the given temperature (Figure 16c). The yield strength sharply decreases at mould operating temperatures above 400 °C, and in some areas, the mould momentarily reaches temperatures up to 600 °C. The presented stress magnitudes (Figure 16) vary during the mould’s operation, from negative (compressive) to positive (tensile) (Figure 16b). Such a cycle of recurring stresses promotes the formation of surface cracks that deepen over time. As noted in [12], the deepest cracks form near the gating system, as confirmed by the numerical calculations (Figure 16c). These operating conditions impose significant demands on the materials and technology used for making the metal mould.

#### 3.2.2. Stress Analysis in a Working Mould with Structural Elements with Simple and Conformal Cooling Made Using Additive Manufacturing from 18Ni300 Powder

Die-casting moulds used in aluminum alloy casting are subjected to varying stress fields, mainly in the surface layers, related to significant temperature gradients resulting from the specified operating cycle of the die-casting machine. The magnitude of these stresses is influenced not only by the number of cycles but also by the thermal load characteristics over time, especially with the use of intensified cooling processes for working elements. A comparison of the stress state during the operation of the die-casting mould was performed on a selected mould element that functions as a metal diffuser to the gating system and then to the mould cavity. This element (Figure 17), due to the moving piston, undergoes high metal pressure of about 70 MPa (700 bar), and then directs this to the gates. Metal flows at speeds ranging from 30 to 40 m/s (110 to 150 km/h). Between the piston front and the diffuser, a solidifying metal layer forms with a diameter of 20 to 100 mm and a thickness of about 25 mm (depending on the type of die-casting machine used). This is a long solidifying area and to speed up the machine cycle, it should be intensively cooled. The dimensions of the diffuser and cooling systems. The simulation setup includes the alloy, which is commonly used DIN 226, the solid elements were modeled in SolidWorks software (https://discover.solidworks.com/contact-a-sales-solidworks?mktid=&utm_campaign=202401_glo_sea_cre_ecal_brand_en_CMP8775_labl&utm_medium=cpc&utm_source=google&utm_content=search&gad_source=1&gclid=CjwKCAiA6aW6BhBqEiwA6KzDc-WbIxPYMUE7HyyPilGimmzBshXiXvElvx4o1PngB26zLMMTwps_DBoC4QgQAvD_BwE (accessed on 25 November 2024)) and uploaded as STL filed for plunger, shot sleeve, die half and plunger movement characteristics for velocity and time. Analysis for the simple cooling system was conducted for the diffuser made of X40CrMoV5_1 steel, while the diffuser with conformal cooling systems was printed from 18Ni300 powder (Figure 18). Water flowing at a rate of 0.5 m^3^/h was used as the coolant in all channels. The heat exchange coefficient between the flowing water and the mould was 8800 W/m²K for the simple channel and 8537 W/m²K for the conformal system.

As expected, the temperature on the surface of the diffuser directly contacting the metal and within its cross-section during the work cycle is lower with the conformal cooling system. With conformal cooling system no. 3, however, the temperature gradient increases both on the surface and in the cross-section of the diffuser. In the graphs (Figure 19, at the marked points P1 and P2 on the surface, the difference averages 165 °C for system no. 3 and about 95 °C for cooling system no. 1. In the cross-section of the diffuser, the average temperature difference between points P1 and P2 is about 140 °C for cooling system no. 3 and 30 °C for cooling system no. 1.

Assessing the stress state throughout the diffuser for all three cooling systems is challenging based solely on selected points from the cross-section or surface of the diffuser. Therefore, in further stress assessment, a bit map image editing method and analysis using the ImageJ program were applied. Bitmap images were generated for individual cooling systems from numerical calculations of selected cross-sections and surfaces directly contacting the metal. The percentage share of Von Mises stresses in cross-sections exceeding 400 MPa and 600 MPa was determined. The adopted 400 MPa threshold represents about 32% of the Rp0.2 strength at 400 °C for both materials, while 600 MPa represents a critical state where stresses approach the Rp0.2 strength at 600 °C. Results of average percentage shares of stresses above 400 MPa in diffusers with analyzed cooling systems indicate similar levels (Figure 20a). Slightly lower percentages occur in the diffuser with the conformal cooling system no. 3 during their maximum values. Maximum stress shares occur in cross-sections where the cooling channel appears (Figure 20b). This is the case in the cross-section at level 0 for cooling system no. 1, where the average value during the process cycle is about 29%, while in the remaining cross-sections, it ranges from 6 to 13%. The percentage share of stresses above 600 MPa on the diffuser surface during operation is significantly lower (Figure 21). The maximum value is about 4% for conformal cooling no. 3. Although the percentage share of stresses above 600 MPa is not large, cyclic and frequent changes occurring every 150 s significantly promote crack formation, especially since this is an area of geometric change ending with a sharp edge. As conformal systems create a larger network of channels, increased oscillation of stress values along the cross-section for the conformal system can be observed. This is very noticeable in the cross-section of the diffuser with conformal cooling system no. 2 compared to the simple cooling system no. 1, where stresses with the simple cooling system are stable at about 500 MPa, while with the conformal system, they vary from 300 to 700 MPa (Figure 22). On the surface of the cooled structural element, compressive stresses usually occur, while tensile stresses are very high near the channels and exceed 600 MPa, whereas compressive stresses on the diffuser surface reach −150 MPa. There is a significant stress gradient between the diffuser surface and the cooling channel. Often, to increase cooling efficiency, the conformal channel systems are placed close to the surface. Such a large gradient over a shorter distance will be a very unfavorable factor and can accelerate the formation of cracks, reducing its lifespan.

Numerical calculations conducted to determine the stress state in a structural element operating in a variable temperature field with cyclic variable loads allowed determining the magnitude and location of their occurrence. Based on conducted research work the diffusor part was designed and printed on the RPMI machine. In Figure 23, the design, printed and machined part is presented.

The presented part is the element of the die-casting mould, which after heat treatment and machining is placed in the machine. Currently, the part is under exploitation since it should be maintained for 500,000 work cycles. 

## 4. Summary and Conclusions

Research has shown that the 3D-printing process using the LMD method results in microporosity within the material structure, which is dependent on printing parameters and powder quality. The lowest porosity was achieved with a laser energy density of 1.717 J/mm^3^ (power of 1300 W, track width of 1.52 mm, layer height of 0.5 mm). The optimal heat treatment parameters for 18Ni300 steel provide an average hardness of 46 ± 2 HRC, a strength of 1600 ± 50 MPa, and an elongation of 10 ± 1.5%. Mechanical tests indicate that the aging temperature significantly impacts the material’s strength and ductility. Higher aging temperatures and longer times increase elongation but decrease strength. The 18Ni300 steel exhibits good strength at elevated temperatures, making it suitable for die-casting moulds. Temperature gradients in pressure moulds generate significant stresses, which are crucial factors affecting mould durability. During the operation of a pressure mould, cyclic temperature changes (from 180 °C to locally 600 °C) cause compressive and tensile stresses, especially near the contact surface with liquid metal. Maximum reduced Von Mises stresses can exceed 400 MPa, and locally even 600 MPa, promoting crack formation. The analysis of the diffusor, from the 18Ni300 powder and equipped with conformal cooling systems, revealed that stresses in the cooling channels are higher than in simple systems. The stress gradient between the diffuser surface and the cooling channel can accelerate crack formation, shortening the die exploitation. Therefore, appropriately designing cooling systems is crucial for minimizing stresses and extending the durability of pressure mould components. Based on the conducted material studies and computer simulations of the diffuser element shape, implementation in foundry conditions is planned. The conducted studies allow for the following conclusions:For the assumed boundary conditions, printing parameters were determined to allow for achieving the minimum porosity in the printed material.Studies on the impact of the heat treatment on the mechanical properties allowed for determining the most favorable process recipe.Numerical analysis of the operation process of the diffuser element, working in a pressure mould, enabled the design of optimal part parameters.Numerical analysis of the stresses arising during operation.

## Figures and Tables

**Figure 1 materials-17-05988-f001:**
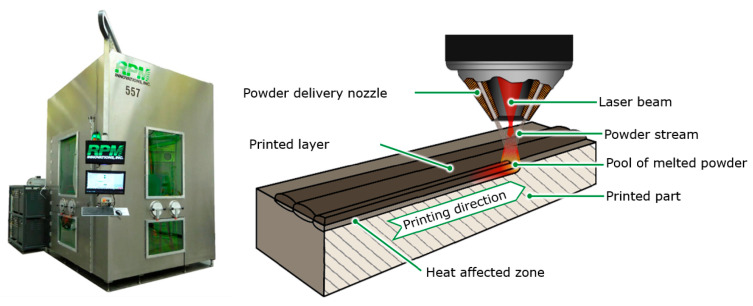
RPMI 557 printer used in the study and the principle of powder feeding and melting using the LMD method.

**Figure 2 materials-17-05988-f002:**
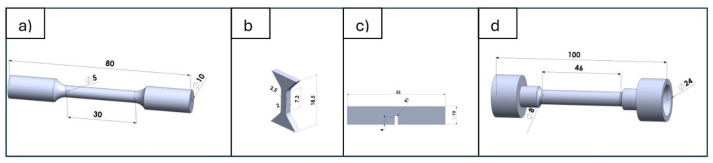
Dimensions and shapes of samples used in the tests: (**a**) static strength at 20 °C, 200 °C, 400 °C, and 600 °C—INSTRON, (**b**) static strength at 20 °C—Zwick, (**c**) impact toughness, (**d**) thermal fatigue.

**Figure 3 materials-17-05988-f003:**
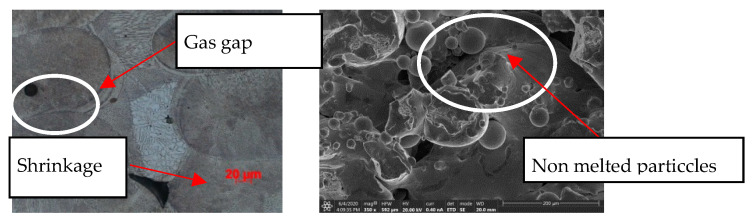
Microstructures of printed samples showing discontinuities in the structure.

**Figure 4 materials-17-05988-f004:**
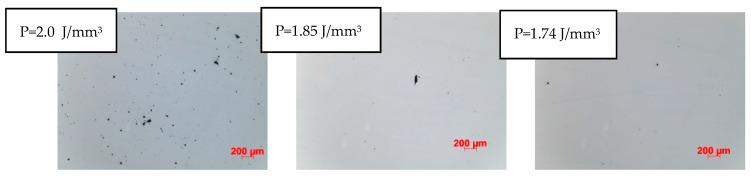
Evaluation of porosity in the metallographic section as a function of laser energy density.

**Figure 5 materials-17-05988-f005:**
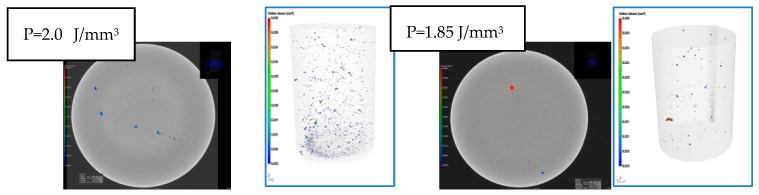
Evaluation of porosity based on CT scan studies.

**Figure 6 materials-17-05988-f006:**
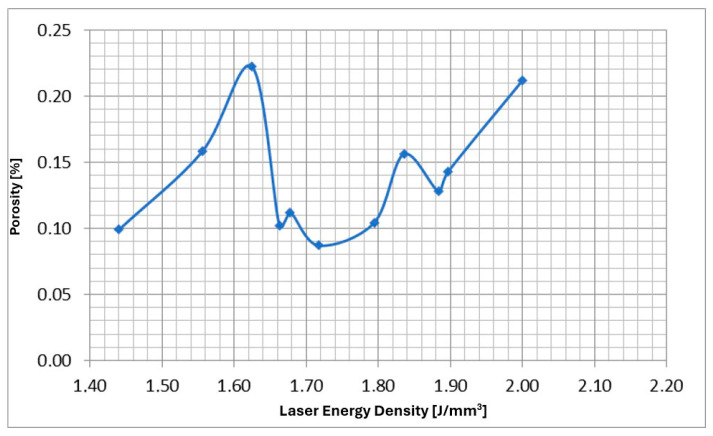
Porosity size as a function of printing parameters expressed by energy density.

**Figure 7 materials-17-05988-f007:**
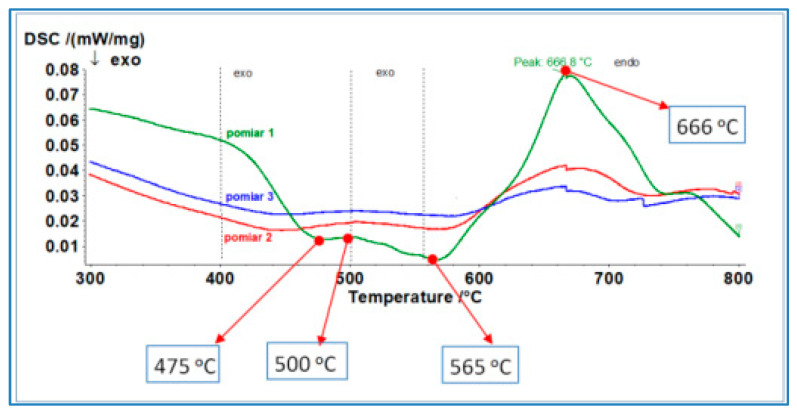
DSC curve for a sample directly printed without heat treatment.

**Figure 8 materials-17-05988-f008:**
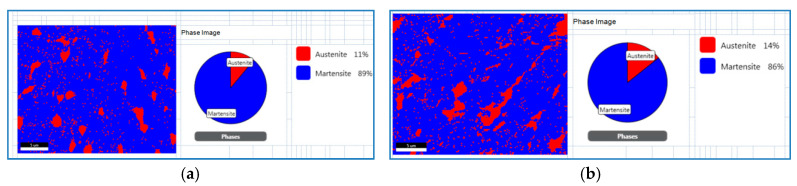
EBSD studies determining austenite content after aging: (**a**) 490 °C for 6 h, (**b**) 560 °C for 8 h.

**Figure 9 materials-17-05988-f009:**
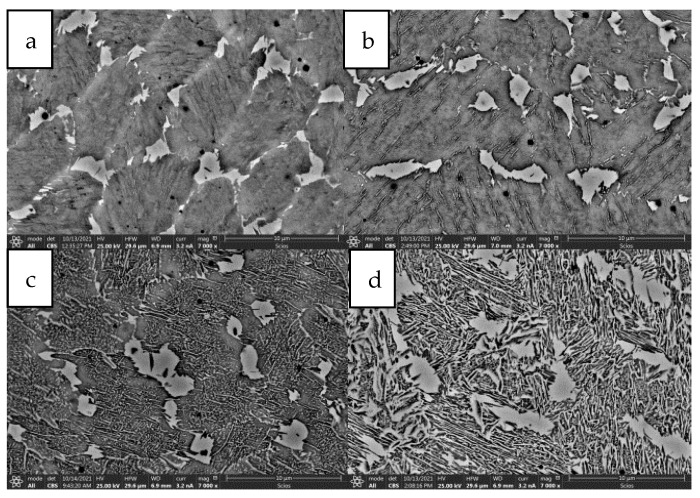
SEM microstructure for heat treatment: (**a**) 490 °C—6 h, (**b**) 540 °C—8 h, (**c**) 560 °C—8 h, (**d**) 600 °C—8 h.

**Figure 10 materials-17-05988-f010:**
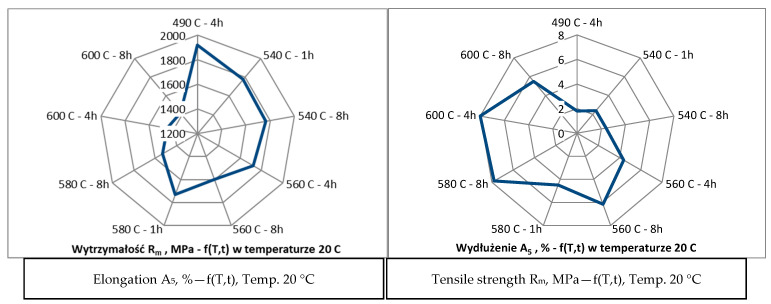
Relationship between tensile strength (R_m_) and elongation (A_5_) at 20 °C.

**Figure 11 materials-17-05988-f011:**
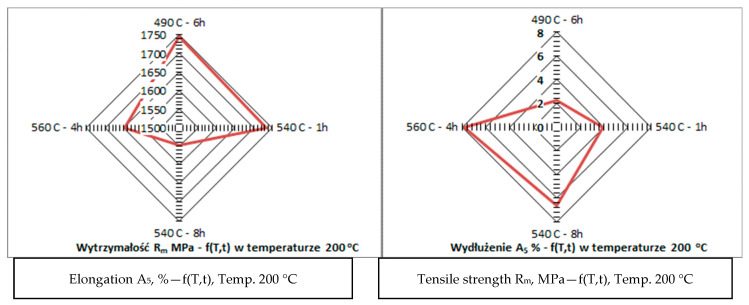
Relationship between tensile strength (R_m_) and elongation (A_5_) at 200 °C.

**Figure 12 materials-17-05988-f012:**
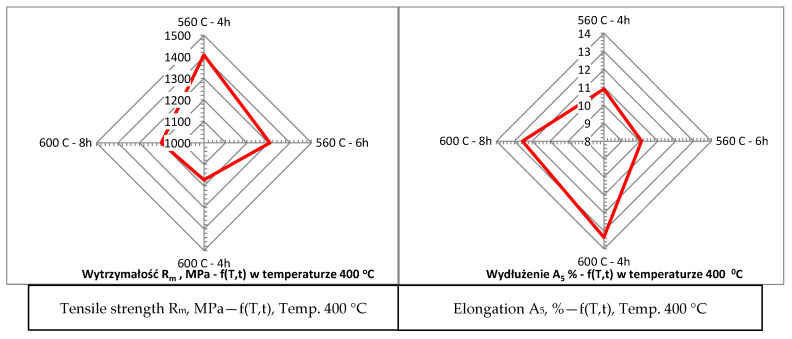
Relationship between tensile strength (R_m_) and elongation (A_5_) at 400 °C.

**Figure 13 materials-17-05988-f013:**
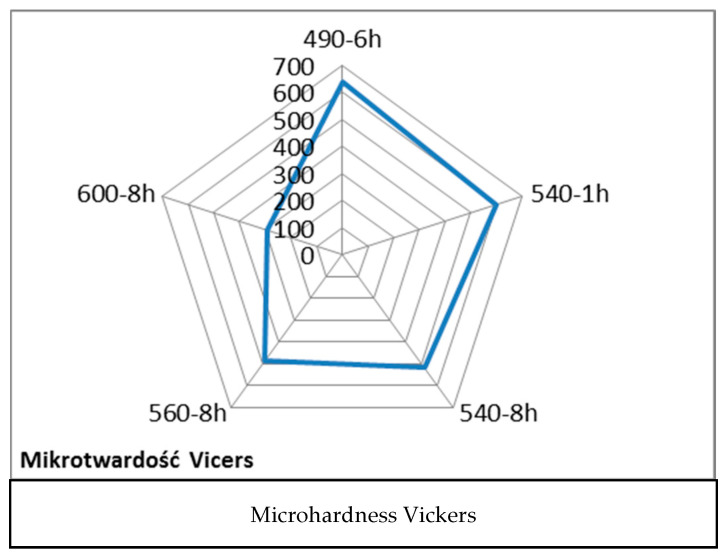
Dependence of tensile strength R_m_ on elongation of samples printed from 18Ni300 powder after heat treatment.

**Figure 14 materials-17-05988-f014:**
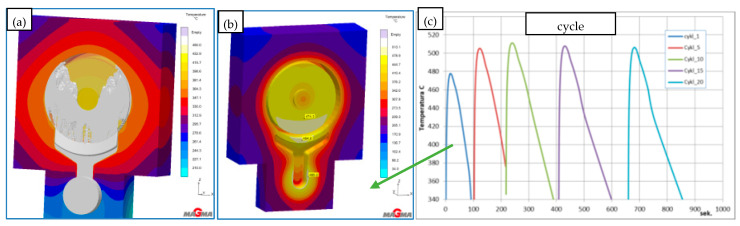
(**a**) Metal flow in the mould and (**b**) temperature distribution at a selected point in the die casting cycle. (**c**) shows graph the temperature distribution.

**Figure 15 materials-17-05988-f015:**
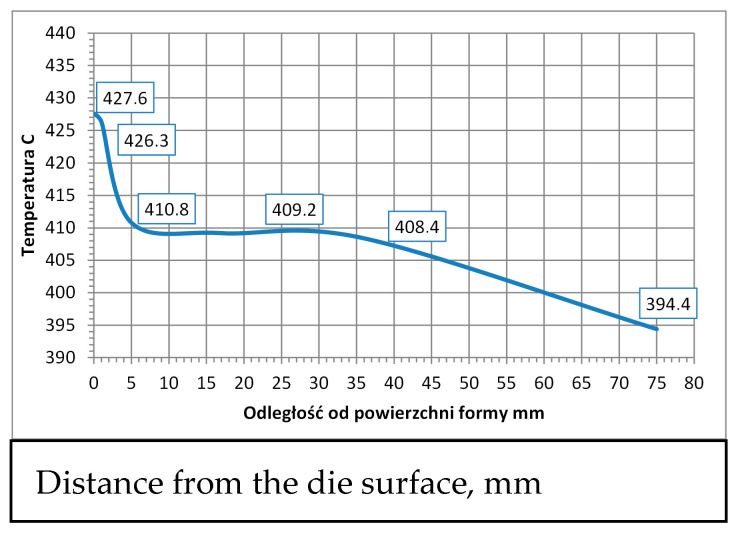
Temperature distribution as a function of distance from the die surface.

**Figure 16 materials-17-05988-f016:**
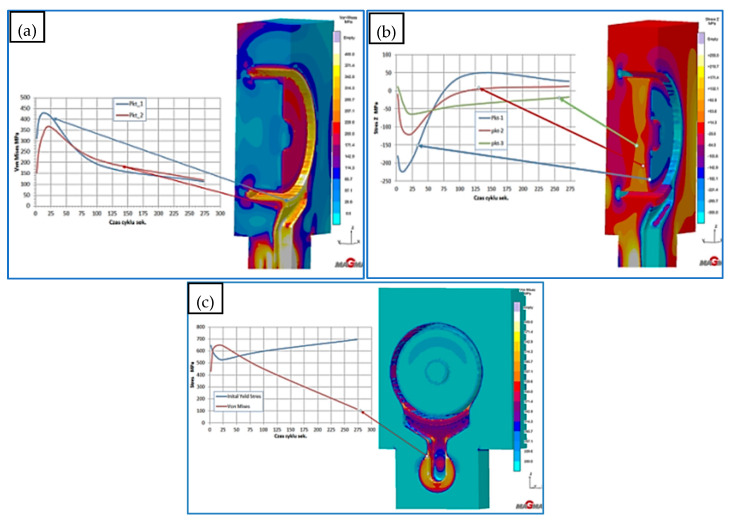
Stress diffuser in a die made of X40CrMoV5_1 steel at selected points: (**a**) reduced Von Mises stresses, (**b**) in the Z-axis direction, (**c**) Von Mises stress magnitudes on the surface relative to the yield strength of X40CrMoV5_1 steel.

**Figure 17 materials-17-05988-f017:**
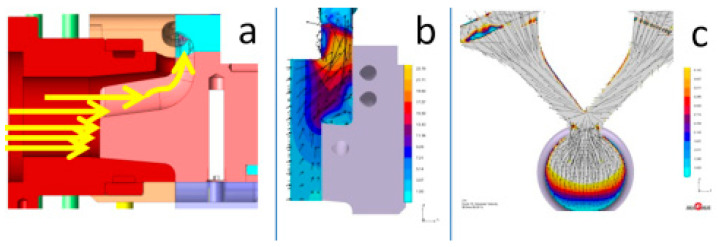
Test mould and flow of the liqiud metal in the die and its direction vectors. (**a**) flow direction in the piston chamber - diffuser system, (**b**) cross-sectional view of the diffuser and flow vectors, (**c**) frontal view of the liquid alloy flow direction.

**Figure 18 materials-17-05988-f018:**
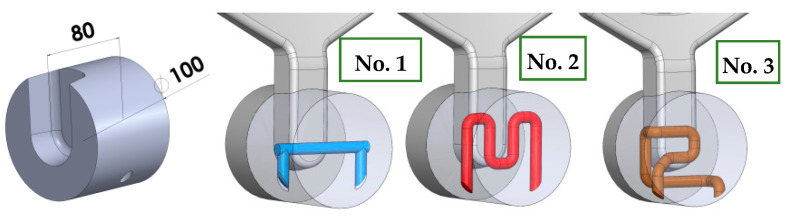
Test metal diffuser subject to analysis with traditional and conformal cooling systems.

**Figure 19 materials-17-05988-f019:**
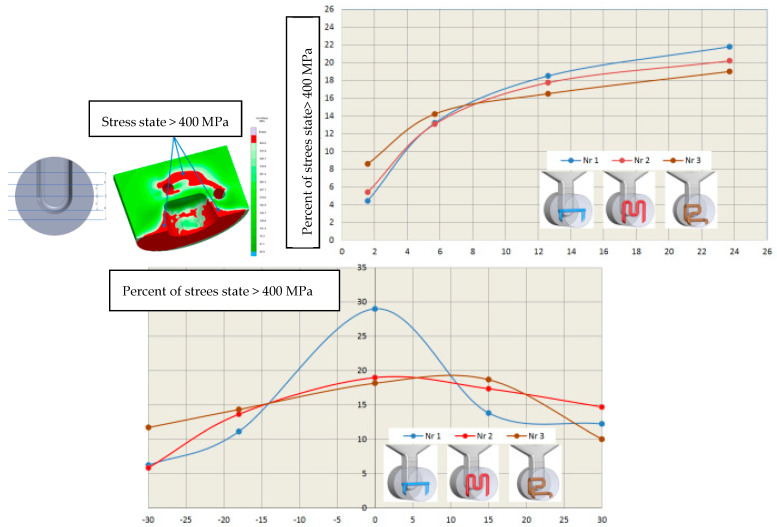
Temperature distribution on the surface and in the cross-section of the diffuser with simple and conformal cooling system no. 3.

**Figure 20 materials-17-05988-f020:**
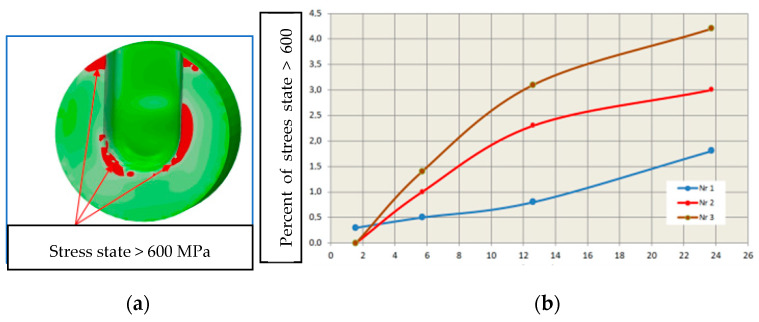
Graphs of average percentage shares of stresses above 400 MPa for three cooling systems (**a**) as a function of time and (**b**) in different diffuser areas throughout the cycle.

**Figure 21 materials-17-05988-f021:**
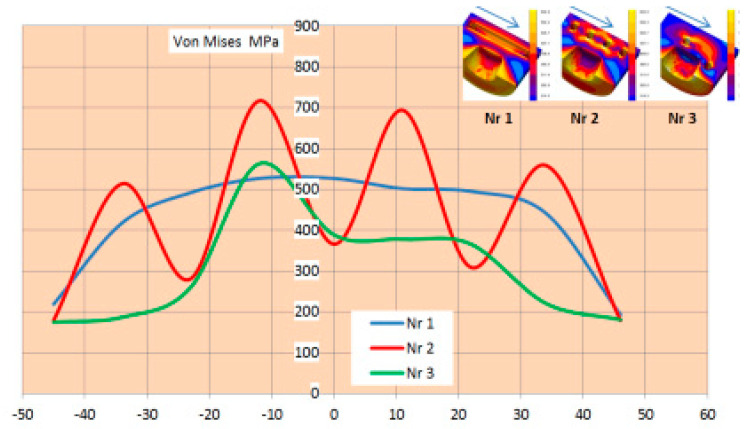
Percentage share of reduced Von Mises stresses above 600 MPa on the diffuser surface.

**Figure 22 materials-17-05988-f022:**
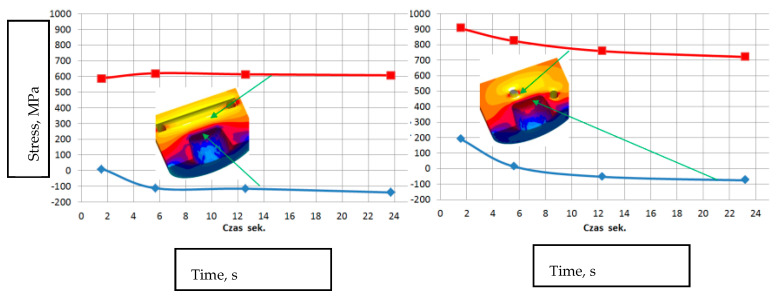
Distribution of reduced Von Mises stresses at 23 s along the cross-section near the cooling channels.

**Figure 23 materials-17-05988-f023:**
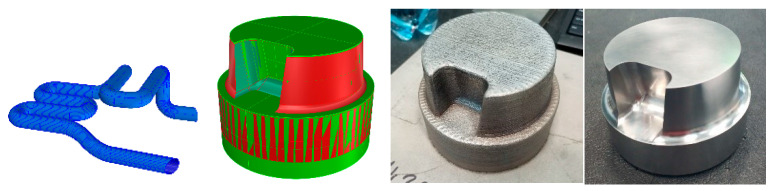
Design of the part for foundry evaluation, cooling channels, 3D scan for dimensional accuracy, printed part and machined part.

**Table 1 materials-17-05988-t001:** Heat treatment properties.

Aging	1	2	3	4	5	6	7	8	9
**Temperature °C**	490	540	540	560	560	580	580	600	600
**Time/h**	6	1	8	4	8	1	8	4	8

**Table 2 materials-17-05988-t002:** Chemical composition of materials used in analysis.

	C	Si	Mn	P	Cr	Mo	V	Co	Cu	Al.	Ti
X40CrMoV5_1	0.4	1.0	0.4	0.035	5.0	1.0	0.3				
Dievar	0.4	0.2	0.5	0.01	5.1	2.2	0.6	0.02	0.05	0.02	0.002

**Table 3 materials-17-05988-t003:** Thermophysical parameters and mechanical properties as a function of temperature.

Temperature (°C)	Thermal Conductivity (W/mK)	Density (kg/m^3^)	Young’s Modulus (MPa)	Thermal Expansion Coefficient (1/°C)	Yield Strength Rp0.2 (MPa)
**X40CrMoV5_1—MAGMA Database**					
20	25.0	7830	216,691	1.03 × 10^−5^	1391
200	27.4	7776	206,100	1.24 × 10^−5^	1366
400	27.3	7711	189,300	1.30 × 10^−5^	1168
600	26.4	7644	165,700	1.00 × 10^−5^	650
**1.2709 maraging steel from 18Ni300 powder—own research**					
20	18.2	7977	213,400	1.03 × 10^−5^	1673
200	13.9	8176	195,900	1.11 × 10^−5^	1463
400	16.9	8078	179,500	1.36 × 10^−5^	1220
600				1.35 × 10^−5^	780

## Data Availability

The original contributions presented in the study are included in the article, further inquiries can be directed to the corresponding author.

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
