# Peer review of "Application of a 3D-Printed Part with Conformal Cooling in High-Pressure Die Casting Mould and Evaluation of Stress State During Exploitation"

_materials, 2024, doi:10.3390/ma17235988_

Round 1
Reviewer 1 Report
Comments and Suggestions for Authors
The article discusses a topical issue related to aluminum alloy casting molds. The issue related to stress formation in the structural elements of a high-pressure casting mold is considered. In this case, the mold elements are manufactured using 3D printing. Before publishing the article, a number of improvements must be made:
1. Increase the number of keywords.
2. Improve the article design. Align paragraphs by width, make indents for the first lines of paragraphs, etc.
3. Improve the Methods and Materials section. It is advisable to provide a general work plan at the beginning of the section. It would also be good to structure the section and introduce subsections for different types of work (Metallography, printing of samples, mechanical testing, ...). For the tests performed, specify in more detail the parameters of the tests performed, the equipment used, and the test modes. Specify the test standards used. For example, for tensile tests, the tensile speed of the samples. Specify the number of samples for each test, the printing speed, the diameter of the printing beam, etc. For the equipment used, indicate the manufacturer and country of manufacture in brackets.
4. Increase the quality of the figures (Figures 7, 8, ...) Enlarge the figures themselves so that the text on them is visible and readable.
5. On lines 250-256, you talk about different amounts of formed austenite. It would be good to provide quantitative data on the amount of this austenite in this analysis. Also describe the amount of other phases, except for austenite, these phases will also affect the properties of the resulting material.
6. In your analysis, you mainly touch on the amount of formed phases. It would be nice to also touch on the issue of the shape of the resulting phases. The shape of phase inclusions also significantly affects the properties of the material.
7. When analyzing porosity, it would be good to consider not only its quantity but also its distribution by volume. To what extent are the pores grouped? How does their shape change, is it always spherical?
8. Add information about the possible practical application of the results of the work obtained to the conclusions.
Author Response
-
- Increase the number of keywords.
Additional key words was added.
- Improve the article design. Align paragraphs by width, make indents for the first lines of paragraphs, etc
The article layout was changed to match the lines and figures.
- Improve the Methods and Materials section. It is advisable to provide a general work plan at the beginning of the section. It would also be good to structure the section and introduce subsections for different types of work (Metallography, printing of samples, mechanical testing, ...). For the tests performed, specify in more detail the parameters of the tests performed, the equipment used, and the test modes. Specify the test standards used. For example, for tensile tests, the tensile speed of the samples. Specify the number of samples for each test, the printing speed, the diameter of the printing beam, etc. For the equipment used, indicate the manufacturer and country of manufacture in brackets.
Methods and Materials section was modified, the addition information regarding procedures was updated.
- Increase the quality of the figures (Figures 7, 8, ...) Enlarge the figures themselves so that the text on them is visible and readable.
Size and quality was changed.
- On lines 250-256, you talk about different amounts of formed austenite. It would be good to provide quantitative data on the amount of this austenite in this analysis. Also describe the amount of other phases, except for austenite, these phases will also affect the properties of the resulting material.
Information regarding microstructure was modified.
- In your analysis, you mainly touch on the amount of formed phases. It would be nice to also touch on the issue of the shape of the resulting phases. The shape of phase inclusions also significantly affects the properties of the material.
In this work, only the fundamental information is presented. A detailed description of the microstructures—including the content of austenite and martensite—will be provided in a separate paper focused on the thermal fatigue resistance of printed components.
- When analysing porosity, it would be good to consider not only its quantity but also its distribution by volume. To what extent are the pores grouped? How does their shape change, is it always spherical?
The work focused solely on analysing the porosity size after printing in relation to the printing parameters. Once optimal data was obtained, no further work was conducted.
- Add information about the possible practical application of the results of the work obtained to the conclusions.
Information was added at the end of results section
Reviewer 2 Report
Comments and Suggestions for Authors
1. The article lacks innovation, and the theoretical analysis is less comprehensive compared to similar studies;
2. The abstract introduction should succinctly outline the main research objectives and experimental findings of the article;
3. The introductory section on the current status of research on the process parameters of mould performance at different temperatures is scarce and should be supplemented;
4. There are many low-level errors in the format of the edition, e.g., should Table 2 and Table 3 be represented in a three-line table? The labeling of figure numbers is also confusing; for example, figures 16, 17, and 18 are not numbered sequentially in the article.
5. The quality of the images in the article is unclear; it is recommended to upload high-resolution images. The number of references is low and additions are recommended.
5.
Comments on the Quality of English LanguageExtensive editing of English language required
Author Response
- The article lacks innovation, and the theoretical analysis is less comprehensive compared to similar studies;
The section was rearranged,
- The abstract introduction should succinctly outline the main research objectives and experimental findings of the article;
The full abstract was rearranged.
- The introductory section on the current status of research on the process parameters of mould performance at different temperatures is scarce and should be supplemented;
Information’s for the section was slightly updated.
- There are many low-level errors in the format of the edition, e.g., should Table 2 and Table 3 be represented in a three-line table? The labelling of figure numbers is also confusing; for example, figures 16, 17, and 18 are not numbered sequentially in the article.
Article format was adjusted. Table 2 and 3 are regarding different topic, so I lest those unchanged.
- The quality of the images in the article is unclear; it is recommended to upload high-resolution images. The number of references is low and additions are recommended.
The quality of the images was enlarged, and changed
Reviewer 3 Report
Comments and Suggestions for Authors
In this paper the authors performed numerical simulations and tests to determine the optimal printing and heat treatment parameters (temperature distribution and stress state ) of 3D printed parts of steel used for manufacturing elements of high-pressure die casting moulds. They used these results as a reference to compare them with the same parts but made from 3D printed maraging steel 18Ni300 powder. They characterized the mechanical and thermophysical properties of the parts also under the application of conformal cooling and found out that the heat treatment is more important for the development of the mechanical properties than the printing parameters. The paper is recommended for publication. A few minor comments follow:
1. The abstract is too long. It should be focused more to what the authors did and the findings of the work rather than detailing the characteristics of the techniques and methods.
2. The introduction reads more like a commentary of the advantages and disadvantages of the additive manufacturing technique and there is almost no information about what the authors did in this work, motivation and novelty of their work related to what has already been done in the literature on the subject.
3. In line 127: is E the scanning energy or energy density? What is the physical meaning of the scanning energy?
4. There is nothing in the text mentioned about Figure 5.
5. How was the line that connects the points in Figure 6 determined? or it is "a guide for the eye"?
6. Have the results in Figure 14 and 145 been taken from the literature or are authors' own work?
7. Figures 19 and 21 should be enlarged and made of higher resolution.
Author Response
-
- The abstract is too long. It should be focused more to what the authors did and the findings of the work rather than detailing the characteristics of the techniques and methods.
The abstract was rearranged.
- The introduction reads more like a commentary of the advantages and disadvantages of the additive manufacturing technique and there is almost no information about what the authors did in this work, motivation and novelty of their work related to what has already been done in the literature on the subject.
In the section some modification has been done, regarding the comments.
- In line 127: is E the scanning energy or energy density? What is the physical meaning of the scanning energy?
Used them was modified.
- There is nothing in the text mentioned about Figure 5.
There are information regarding the fig. 5 in the text.
- How was the line that connects the points in Figure 6 determined? or it is "a guide for the eye"?
More "a guide for the eye" than any other reason.
- Have the results in Figure 14 and 145 been taken from the literature or are authors' own work?
The section was modified.
- Figures 19 and 21 should be enlarged and made of higher resolution.
Figures was changed.
Round 2
Reviewer 1 Report
Comments and Suggestions for Authors
The authors have done a very good job on the article. In my opinion, the article can be accepted.
Author Response
The authors have done a very good job on the article. In my opinion, the article can be accepted.
Thank you.
Reviewer 2 Report
Comments and Suggestions for Authors
1. The treatment of the pictures in the article is slightly rough, please redraw them, as Fig.2.、Fig.4.、Fig.8.、Fig.9.、Fig.14.、Fig.16.、Fig.18.
2. It is best to have a consistent format for the table, such as Table 1.
3. There are certain formatting and grammatical errors in the article, please proofread carefully.
4. There are few references, and there are formatting issues with its citations.
Comments on the Quality of English Language3. There are certain formatting and grammatical errors in the article, please proofread carefully.
Author Response
- The treatment of the pictures in the article is slightly rough, please redraw them, as Fig.2.、Fig.4.、Fig.8.、Fig.9.、Fig.14.、Fig.16.、Fig.18.
Figures have been modified as much as possible. - It is best to have a consistent format for the table, such as Table 1.
The appearance of the tables is uniform - There are certain formatting and grammatical errors in the article, please proofread carefully.
The text has been modified - There are few references, and there are formatting issues with its citations.
Section was modyfied